# Investigation of an Alternative Marker for Hypermutability Evaluation in Different Tumors

**DOI:** 10.3390/genes12020197

**Published:** 2021-01-29

**Authors:** Anqi Chen, Suhua Zhang, Lei Xiong, Shihan Xi, Ruiyang Tao, Chong Chen, Jixi Li, Jinzhong Chen, Chengtao Li

**Affiliations:** 1Department of Forensic Medicine, School of Basic Medical Sciences, Shanghai Medical College, Fudan University, Shanghai 200032, China; 19111010078@fudan.edu.cn; 2Shanghai Key Laboratory of Forensic Medicine, Shanghai Forensic Service Platform, Academy of Forensic Science, Ministry of Justice, Shanghai 200063, China; zsh-daisy@163.com (S.Z.); xionglei13142020@163.com (L.X.); xishihan2020@163.com (S.X.); taoruiyang163@163.com (R.T.); 18883368974@163.com (C.C.); 3State Key Laboratory of Genetic Engineering, Institute of Genetics, School of Life Sciences, Fudan University, Shanghai 200433, China; lijixi@fudan.edu.cn

**Keywords:** microsatellite instability (MSI), short tandem repeats (STRs), tumors, hypermutability

## Abstract

A growing number of studies have shown immunotherapy to be a promising treatment strategy for several types of cancer. Short tandem repeats (STRs) have been proven to be alternative markers for the evaluation of hypermutability in gastrointestinal (GI) cancers. However, the status of STRs and microsatellite instability (MSI) in other tumors have not yet been investigated. To further compare STR and MSI alterations in different tumors, a total of 407 paired DNAs were analyzed from the following eight tumor types: breast cancer (BC), hepatocellular cancer (HCC), pancreatic cancer (PC), colorectal cancer (CRC), gastric cancer (GC), lung cancer (LC), esophageal cancer (EC), and renal cell cancer (RCC). The STR alteration frequencies varied in different tumors as expected. Interestingly, none of the patients possessed MSI-low (MSI-L) or MSI-high (MSI-H), except for the GI patients. The highest STR alteration was detected in EC (77.78%), followed by CRC (69.77%), HCC (63.33%), GC (54.55%), LC (48.00%), RCC (40.91%), BC (36.11%), and PC (25.71%). The potential cutoff for hypermutability was predicted using the published objective response rate (ORR), and the cutoff of LC and HCC was the same as that of GI cancers (26.32%). The cutoffs of 31.58% and 10.53% should be selected for BC and RCC, respectively. In summary, we compared MSI and STR status in eight tumor types, and predicted the potential threshold for hypermutability of BC, HCC, CRC, GC, LC, EC, and RCC.

## 1. Introduction

Microsatellite instability (MSI) has played an essential role in tumor research [1]. In 2017, programmed death 1 (PD-1) inhibitor was approved by the Food and Drug Administration (FDA) for patients with microsatellite instability-high (MSI-H) or mismatch repair protein deficiency (MMR-D) solid tumors, regardless of tumor site or histology [2]. To screen MSI-H- or MMR-D-positive individuals, several methods have been generated, such as the commonly used mono- and di-nucleotide repeats [3], elevated microsatellite alterations at selected tetranucleotide repeats (EMAST) [4,5], and the next-generation sequencing (NGS)-based tumor mutation burden (TMB) [6]. The overexpression of PD-L1 should enrich the response of PD-1 inhibitors; however, samples deemed to be PD-L1-positive do not always respond to immunotherapy [7]. A study with a large sample size showed that only 5–21% of patients suffering from gastrointestinal cancers (GI) comprised MSI-H, which was far from the actual immune check point inhibitors’ response rate (~30%) [8]. It is well known that recovery is more likely if the disease is treated at an early stage. Therefore, more sensitive methods for screening MSI-H individuals is necessary.

Previously, we generated a method, using short tandem repeat (STR) markers, for screening patients who might possess hypermutability. However, the sample types were limited to colorectal cancer (CRC) and gastric cancer (GC) only [9], and further studies on STR status in other type of tumors are needed. Worldwide, MSI status evaluation has been routinely tested in CRC and is now considered a generalized cancer phenotype [10,11]. Many studies have been carried out for MSI evaluation in both colorectal [12] and non-colorectal cancer [13], which have included NGS-based large-scale analysis, PCR-based microsatellite detection, and screening for loss of MMR protein expression using immunohistochemistry [10]. In our previous study [9], we investigated the MSI status of GI using six microsatellite markers, but performance in other tumor samples remains unknown. Therefore, evaluating MSI status in other tumor types is also important.

In the present study, we compared MSI and STR alterations from tumors of breast cancer (BC), hepatocellular cancer (HCC), pancreatic cancer (PC), lung cancer (LC), esophageal cancer (EC), renal cell cancer (RCC), CRC, and GC, and then we predicted the potential threshold of hypermutability for each of the tumors. The study is important to fill gaps in knowledge of STR applicability in other tumors, and to provide further opportunities to use immune check point inhibitors.

## 2. Materials and Methods

### 2.1. Patients and Samples

A total of 407 paired samples were obtained from 36 BC samples, 30 HCC samples, 35 PC samples, 129 CRC samples, 121 GC samples, 26 LC samples, 8 EC samples, and 22 RCC samples. The patients underwent surgical tumor resection in 2013–2019 at the Changhai Hospital, Second Military Medical University, and the Shanghai and Huadong Hospital Affiliated with Fudan University, Shanghai. All samples were collected upon the approval of the Ethics Committee of Academy of Forensic Science, Ministry of Justice, China (No. SJY2013-W002, approved 4 January 2013). All participants provided written informed consent.

Tissue samples were obtained from resected tumors. Para-carcinoma or peripheral blood was used for control DNA isolation. The relative percentage of tumor cells to nucleated cells was assessed by a senior pathologist after hematoxylin and eosin staining. Samples with at least 30% tumor cells were considered for further study.

### 2.2. DNA Preparation

Tumor tissues from 10 formalin-fixed paraffin-embedded (FFPE) slides were extracted using a QIAamp DNA FFPE Tissue Kit (Qiagen, Valencia, CA, USA). Blood control DNA was extracted from 100 μL of peripheral blood using a QIAamp DNA Blood Kit (Qiagen, Venlo, The Netherlands). All DNA was extracted in accordance with the manufacturer’s instructions and quantified using a Qubit fluorometer (Life Technologies, Carlsbad, CA, USA). Extracted DNA was stored at –80 °C until use.

### 2.3. Evaluation of Microsatellite Instablity Stability and Short Tandem Repeats Alteration Status

Microsatellite instability stability of the paired DNAs was evaluated using an MSI Detection Kit (Microread, Beijing, China). The kit includes six quasimonomorphic mononucleotide repeats (BAT-25, BAT-26, NR-21, NR-22, NR-24, and MONO-27), two STRs (Penta C and Penta D), and a sex-related polymorphism, Amel. The MSI status was evaluated in accordance with the manufacturer’s instructions by comparing the matching normal and tumor sample pairs for shifts in allele sizes. Generally, the tumor is considered instability-high (MSI-H) if more than 30% (2/6) of the loci are altered and instability-low (MSI-L) if the loci are altered by less than 20% (1/6). MSS refers to the samples with stable microsatellites.

Short tandem repeats were determined using either the Goldeneye^®^20A Forensic Identifier Kit (Peoplespot, Beijing, China) comprising 20 STRs or the SiFaSTR^TM^ 23-plex system comprising 23 STRs. Fluorescent multiplex polymerase chain reaction (PCR) was used in accordance with the manufacturer’s protocol. Genotyping was performed in a 3130xl ABI Prism Genetic Analyzer (Applied Biosystems, Waltham, MA, USA) using GeneMapper Software (Applied Biosystems, Waltham, MA, USA). STR status was classified with the paired samples by investigating for the detection of genotypes among the 19 somatic STR markers (CSF1PO, D12S391, D13S317, D16S539, D18S51, D19S433, D21S11, D2S1338, D3S1358, D5S818, D6S1043, D7S820, D8S1179, FGA, Penta D, Penta E, TH01, TPOX, and vWA). Against the control STR type, three types of STR alterations were determined and calculated for the respective samples, as previously mentioned [9], namely allelic loss (L), occurrence of an additional allele (Aadd), and occurrence of a new allele (Anew). Briefly, L was defined when the peak ratio in the tumor sample/corresponding peak ratio in control blood was <0.5 or >2. Aadd was defined when an additional allele occurred in the tumor sample (e.g., allele 16, 17 > allele 16, 17, 19). Anew was defined when the allele replacement occurred in the tumor sample (e.g., allele 16, 17 > allele 16, 19).

### 2.4. Statistics Analysis

Statistics were performed using Prism 4.0 software (GraphPad, San Diego, CA, USA).

## 3. Results

### 3.1. Widespread STR Alterations Observed across the Tumors

The 407 paired tissues from eight types of tumor were collected for the study, which revealed a varied STR mutation frequency of 5.12% ± 4.06% to 25.73% ± 11.43% in the different tumors. EC exhibited an alteration frequency of 25.73% ± 11.43%, which was the highest compared to the rest of the tumor types. CRC was next highest with a 16.85% mutation rate, followed by GC (14.97%) and LC (13.48%). Comparatively low alteration rates were shown in BC (5.12% ± 4.06%), HCC (10.88% ± 10.70%), PC (5.67% ± 2.87%), and RCC (6.05% ± 4.80%) (Figure 1).

### 3.2. Loss of Heterozygosity Was the Most Commonly Observed Alteration in Eight Types of Tumor

Three types of STR alterations were defined, as previously reported [9]. In the present study, a total of 1016 STR alterations were observed in the 407 paired tumor samples. As shown in Figure 2A, the occurrences of L, Aadd, and Anew were 78.54%, 17.81%, and 3.64%, respectively. The number of alterations of L was decidedly in the majority regardless of tumor type and STR loci, and of all types of alterations, those of Anew were seldom detected. More Aadd alterations were observed in CRC and GC (Figure 2B,C).

### 3.3. Different Tumors Exhibited Varied Alteration Tendencies in the Loci

There were eight types of tumor collected in the study, and the alterations were distributed differently across the tumors. As shown in Table 1, the landscapes in alteration frequency were different, and no particular regularity could be found except for the lowest alteration rate of TPOX (4.34% ± 3.55%). D13S317 was the most frequently (19.86%) altered locus in the study, followed by FGA (16.62%), D18S51 (16.77%), D6S1043 (15.49%), and D8S1179 (14.99%).

Wide ranges of STR alterations were detected in CRC, GC, EC, and LC, while the STRs were selectively altered in BC, HCC, PC, and RCC. The standard deviation for each of the loci ranged from 3.55% to 15.36%, indicating the inclined alteration in STRs. For example, mutations of D19S433, D7S820, and CSF1PO were frequently altered in CRC, GC, EC, and LC. D6S1043, FGA, D16S539, and D13S317 were extremely high in HCC, and D18S51 possessed the highest alteration rate in CRC, which demonstrated a potential hotspot for the corresponding tumors.

### 3.4. MSI Status Was Only Detected in Gastrointestinal Tumors

A total of 84 MSI alterations were detected in the 407 paired tumors, and MSI status of each of the samples was evaluated by MSI-L, MSI-H, or MSS. Interestingly, all of the samples exhibited MSS, except for those from the GC and CRC groups. The CRC samples of 4.65% possessed MSI-H, and 0.78% of them were evaluated as MSI-L. The MSI status in GC was similar to that observed in CRC, namely that most of the samples were MSS. The percentage of MSI-H and MSI-L was 7.44% and 2.48%, respectively, in GC. Among the 84 alterations, there were 31 MSI alterations observed in CRC, and 53 alterations were detected in GC. The most frequently altered MSI locus in GI was BAT25 (Table 2).

### 3.5. Comparison between the Alterations of STR and MSI

Both the STRs and MSI were tested for all 407 paired samples, and more STR alterations than MSI alterations were detected. As shown in Table 3, the STR alteration frequency of the MSI-H samples ranged from 26.32% to 84.21%, and that of the MSI-L samples ranged from 31.58% to 73.68%. Therefore, all MSI-H samples possessed varied degrees of STR mutations. As for the MSS cases, the STR alteration frequency was in the range of 0–66.67% (Appendix A), which indicated that the STRs showed more alteration than did the MSI.

## 4. Discussion

Mutations in somatic cells can be induced spontaneously or via an environmental burden, which can lead to the occurrence of tumorigenesis [8]. Detection of MSI-H has been an effective and robust method for the evaluation of tumor hypermutability, which is considered to be a predictive biomarker for the therapeutic response of an immune checkpoint blockade [2]. Nowadays, TMB is widely recognized as a biomarker for immunotherapy response [16]. Use of immune checkpoint blockades in patients with GI has a proven response rate of 30% in the clinic, while only a few of them (less than 30%) are screened as MSI-H [8]. The structure of MSI and STRs are both repeated nucleotides unit, and thus they may share the same heredity origin and may be useful for hypermutability evaluation. In our previous study, using a forensic STR kit, we found that more alterations could be detected in STRs compared to conventional MSI detection, which indicated more potential immunotherapy beneficiaries in CRC and GC [9]. However, the tumor types were limited to only GI tumors, and the alteration status in other major types of cancers remained unknown. To fill gaps in our knowledge, samples from eight types of tumors were recruited and used for the analyses of STRs and MSI.

The landscapes of STR alterations from different tumors were analyzed firstly. The STR status of all the 407 paired samples was evaluated using the 19 STR loci. All STR alterations mentioned by Chen et al. [9] were observed in the present study. Allelic loss (L) remained the most frequently altered mutational type compared to the others, and the occurrence of an additional allele (Aadd) ranked second, regardless of tumor type or STR loci (Figure 2). The result was concordant with data published previously [14,17,18]. STR genotyping plays an important role in forensic identification, but any alterations may lead to misinterpretation of the results [19]. As for the evaluation of tumor hypermutability, STR alteration was measured by the counts of the mutational loci. Therefore, the alteration type of the loci was not so important. Widespread allelic alterations were detected across the different loci and tumors, and all mutation frequencies were higher than their germlines’ mutation rates (Table 1). The observations indicated that STRs may be potential markers to evaluate high or low tumor mutation burden. The highest genetic alteration frequency was observed in EC, while the alteration rates in RCC, PC, and BC were comparative lower. Chalmers et al. [20] demonstrated, using comprehensive genomic profiling, that TMB is strongly reflective of that from whole exome sequencing. We compared TMB results from published manuscripts to the STR alteration frequencies, since STR alterations and TMB all both markers used to predict hypermutability [9]. Although the calculations for the two methods are different, the trends of mutability were similar for each (Figure 3). In addition, the values of STR alterations were higher than those of TMB, which again demonstrated that the STRs might be a more sensitive predictor of hypermutability.

Short tandem repeats are usually located in non-transcriptional regions, such as intergenic regions and introns. Previous studies [25] demonstrated that STR replacements in introns played an important role in gene expression and disease, which included several well-known forensic STRs. For example, STR replacement in vWA may lead to hemophilia and menorrhagia [26]. We also tried to search the location of each STR locus to determine the potential clinical relevance. Among the 19 targeted STRs, there were eight (D6S1043, D5S818, D21S11, D18S51, D16S539, Penta E, D13S317, and D12S391) located in regions with no genes, and the others were all located in gene introns (NCBI Genome Data Viewer). D19S433 was found to be located in the noncoding sections of unconventional prefoldin RPB5 interactor 1 (URI), which is a component of the PAQosome (Particle for Arrangement of Quaternary structure). URI was predicted to regulate R2TP complex activity and to be involved in localization, stabilization of RPB5, and transcriptional regulation [27]. Chaves-Pérez et al. [28] showed that overexpressed URI in the intestine protected mice from radiation-induced gastrointestinal syndrome (GIS), while URI1-overexpressed tumors displayed decreased transcription levels of tumor suppressor in uterine carcinosarcoma [29]. In the present study, alterations in D19S433 were comparatively higher in CRC (14.73%) and GC (14.05%). Based on the conclusions mentioned above, D19S433 may lead to abnormal URI expression in GI. However, the hypothesis should be illustrated using experimental proofs. HCC possessed a low alteration rate compared to the others, except for D6S1043, FGA, D16S539, and D13S317. FGA was found to have significantly different expression profiles of plasma samples from HCV-infected alcoholic patients [30], and may play a role in premalignant and precancerous lesions of HCC [31]. Although D6S1043, D16S539, and D13S317 were also frequently altered in HCC, correlations could not be generated, as no genes were observed in the region. Taken together, these findings may represent potential mutational hotspots for further tumor research, as the molecular mechanisms responsible for the tumors remain largely unknown.

Microsatellite instability detection plays an essential role in tumor research [32], as does MSI status evaluation for each tumor. However, neither MSI-H or MSI-L were observed in the tumor, regardless of GI cancer type (Table 2). The observation was in concordance with clinical experiences, where most blockades of immune checkpoints have been used for the treatment of GI cancers [33,34]. It has been widely understood to date that MSI-H patients with GI cancers may benefit from immunotherapy [35], although the MSI status for other types of tumors has been less reported. To screen MSI-H individuals, the National Cancer Institute (NCI) has recommended five microsatellite markers for detection [36]. A six-MSI marker system was applied in the present study. Based on the MSI platform, the patients may harbor mutations of 16.67% (1/6), 33.33% (2/6), 50% (3/6), or more. Since MSI-H and MSI-L are defined as patients who harbor one or more than one alteration, respectively, the interpretation for patients harboring 16.67–33.33% alteration is ambiguous. According to a survey in 27 types of cancers, the response rate of PD-1 and PD-L1 for GI cancers was 30% [8], while only 5–20% of MSI-H patients could be screened in clinics [37,38,39]. Laiho et al. [40] surmised that all CRCs would possess some degree of MSI if enough markers were tested. With the development of precision medicine, a larger number of markers could be tested using high-throughput methods (e.g., NGS) [41]. However, the NGS service has not been easy to popularize because of its high expense and need for professional expertise. STR alterations have proven to be alternative markers for MSI with a threshold of 26.32% as the potential cutoff for hypermutability in GI cancers [9]. In the present study, we calculated the number of samples that possessed STR alterations as more than 26.32%. It seems that the value of 26.32% remained the closest cutoff to the expected response rate for LC and HCC, while higher STR alteration frequency should be selected for BC (≥31.58%) and EC (≥47.37%) to meet the desired objective response rate (Table 4). There was a total of nine STR positive samples detected in RCC, and the potential cutoff value should be settled to ensure ~5 samples are being theoretically screened out. Therefore, the cutoff value for RCC should be 10.53%. As for pancreatic cancer, although STR alterations were detected in the study, no ORR data from large sample sized studies have been released so far. One study using a small number of samples revealed that the ORR of immune checkpoint inhibitors combined with chemotherapy was 18.2% [42]. Therefore, the cutoff for PC is not predictable under the current situation.

## 5. Conclusions

Cancer is the leading cause of death worldwide, and immunotherapy has proved to be a remarkable drug for MSI-H and TMB-H patients [43]. Therefore, developing new methods to screen patients who might benefit from immune checkpoint inhibitors is important. Previous studies have shown that STRs could be an alternative marker for hypermutability evaluation in GI cancers, while the applicability in other tumors has remained unknown. In the present study, the alteration status in different tumor types was investigated. Higher STR alteration frequency was observed compared to that of the MSI, which indicated the STRs might also be a more sensitive bio-marker for other tumor types. In the present study, no MSI positive sample was observed regardless of GI tumors. However, Zhang et al. [44] found the ratios of MSI-H in different types of solid tumors were around 0.5~2%. It seemed that the sampling error cannot be completely avoided in clinical practice. Lin et al. [45] once investigated the MSI status in LC patients, and observed only 0.21% (12 out of 5592) of them exhibited the characteristic of MSI-H. Compared to the article mentioned above [44,45], our results were achieved based on a limited sample size, which might lead to biased observations. The best way to measure the diagnostic value and effectiveness of STR markers is through clinical outcomes. Therefore, future studies should use larger sample sizes, and STR markers should be validated with clinical follow-up. In summary, STRs were found to be an alternative marker for hypermutability evaluation, but more clinic trials should be followed to verify the observations.

## Figures and Tables

**Figure 1 genes-12-00197-f001:**
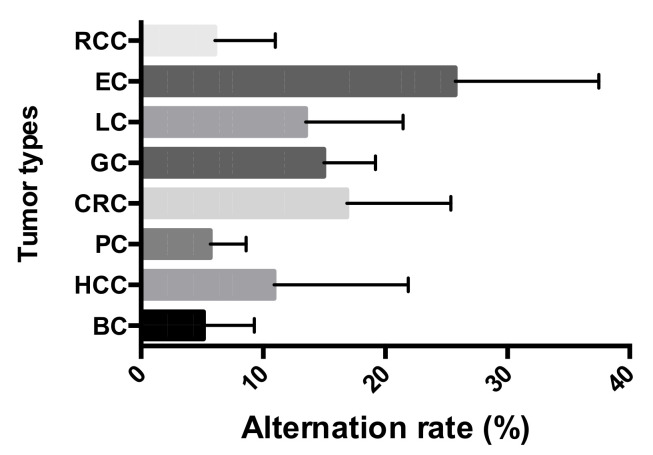
Short tandem repeat (STR) alteration frequencies across the eight tumor types: Breast cancer (BC), hepatocellular cancer (HCC), pancreatic cancer (PC), colorectal cancer (CRC), gastric cancer (GC), lung cancer (LC), esophageal cancer (EC), and renal cell cancer (RCC).

**Figure 2 genes-12-00197-f002:**
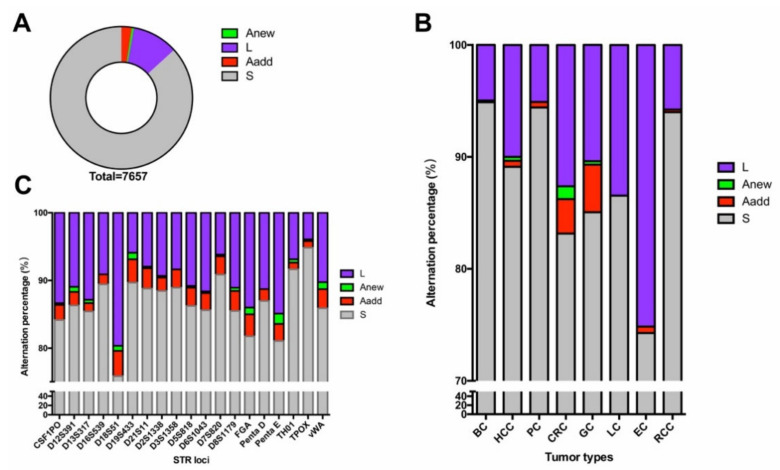
STR alterations across the paired tumors: (**A**) Distribution of the genetic instabilities in STR mutations, (**B**) frequency of genetic status at each of the tumor type, (**C**) percentage of genetic status at each of the STR locus. Occurrence of a new allele (Anew), loss the heterozygosity (L), occurrence of an additional allele (Aadd), stable (S).

**Figure 3 genes-12-00197-f003:**
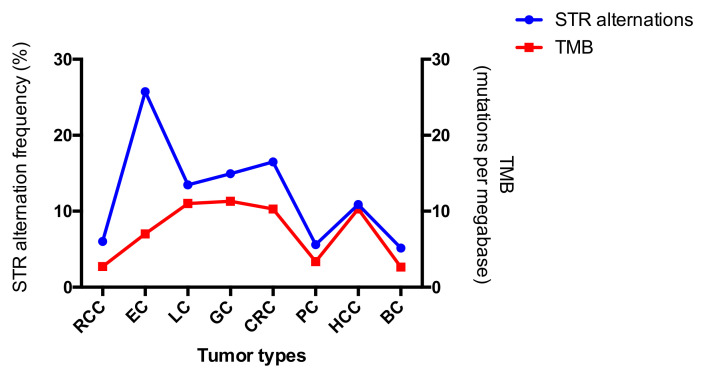
Comparison of tumor mutation burden (TMB) and STR alteration across the eight tumor types. The corresponding TMB was derived from data from Barroso-Sousa et al. [21], Chen et al. [22], Wang et al. [23], Ji et al. [24], and Chalmers et al. [20].

**Table 1 genes-12-00197-t001:** STR alterations in each of the loci in the 407 paired tumor samples. The background color of green indicated low mutation frequency, and it turned to red when the mutation frequency grew higher.

Locus	Germline Mutation Frequency [14,15]	BC	HCC	PC	CRC	GC	LC	EC	RCC
D19S433	0.11%	0.00%	0.00%	2.86%	14.73%	14.05%	8.00%	33.33%	0.00%
D7S820	0.10%	2.78%	0.00%	2.94%	10.08%	14.05%	8.33%	22.22%	4.55%
D6S1043	0.14%	2.78%	26.67%	7.14%	13.18%	17.50%	8.33%	33.33%	15.00%
CSF1PO	0.16%	2.78%	3.33%	6.25%	23.26%	21.49%	12.00%	11.11%	0.00%
D5S818	0.11%	0.00%	0.00%	8.57%	24.81%	12.40%	20.00%	11.11%	0.00%
FGA	0.28%	2.78%	30.00%	6.45%	24.03%	19.33%	12.50%	33.33%	4.55%
D3S1358	0.12%	5.56%	10.00%	5.71%	9.30%	12.40%	16.00%	44.44%	13.64%
D2S1338	0.12%	2.78%	6.67%	2.86%	13.18%	15.70%	16.00%	22.22%	4.55%
TPOX	0.01%	2.78%	3.33%	0.00%	7.75%	5.79%	4.00%	11.11%	0.00%
D21S11	0.19%	0.00%	3.33%	6.45%	14.73%	10.74%	20.83%	33.33%	9.09%
Penta D	0.14%	0.00%	3.33%	6.45%	15.50%	16.81%	16.67%	22.22%	13.64%
D18S51	0.22%	5.56%	6.67%	9.38%	43.41%	21.85%	16.00%	22.22%	9.09%
D16S539	0.11%	11.11%	23.33%	2.86%	9.30%	14.88%	4.00%	0.00%	0.00%
Penta E	0.16%	11.11%	6.67%	7.14%	26.56%	19.33%	20.83%	33.33%	9.09%
D13S317	0.14%	11.11%	36.67%	2.94%	13.28%	9.92%	36.00%	44.44%	4.55%
D12S391	0.24%	8.33%	13.33%	9.68%	13.95%	16.53%	16.67%	22.22%	4.55%
Vwa	0.17%	8.33%	13.33%	11.43%	16.67%	16.81%	4.00%	33.33%	4.55%
TH01	0.01%	8.33%	3.33%	2.86%	10.08%	9.09%	4.00%	22.22%	9.09%
D8S1179	0.14%	11.11%	16.67%	5.71%	16.28%	15.70%	12.00%	33.33%	9.09%
Average	☐	5.12%	10.88%	5.67%	16.85%	14.97%	13.48%	25.73%	6.05%
Standard deviation	☐	4.06%	10.70%	2.87%	8.28%	4.11%	7.76%	11.43%	4.80%

The background color of green indicated low mutation frequency, and it turned to red when the mutation frequency grew higher. **☐**, It was the empty colomns.

**Table 2 genes-12-00197-t002:** Microsatellite instability (MSI) status of the 407 paired cancer samples across the eight tumor types. S refers to stable, which refers to the locus without alterations; I is short for instable, which refers to alterations in the locus. BC (*n* = 36); HCC (*n* = 30); PC (*n* = 35); CRC (*n* = 129); GC (*n* = 121); LC (*n* = 26); EC (*n* = 8); RCC (*n* = 22).

Tumor Type	No. of Loci	MSI Loci						MSI Status		
		NR21	BAT26	NR27	BAT25	NR24	MONO27	MSS	MSI-L	MSI-H
BC	S	36	36	36	36	36	36	36 (100%)	0 (0%)	0 (0%)
	I	0	0	0	0	0	0			
HCC	S	30	30	30	30	30	30	30 (100%)	0 (0%)	0 (0%)
	I	0	0	0	0	0	0			
PC	S	35	35	35	35	35	35	35 (100%)	0 (0%)	0 (0%)
	I	0	0	0	0	0	0			
CRC	S	124	123	123	122	124	125	122 (94.57%)	1 (0.78%)	6 (4.65%)
	I	5	6	6	7	5	4			
GC	S	113	113	113	110	114	112	109 (90.08%)	3 (2.48%)	9 (7.44%)
	I	8	8	8	11	7	9			
LC	S	25	25	25	25	25	25	25 (100%)	0 (0%)	0 (0%)
	I	0	0	0	0	0	0			
EC	S	9	9	9	9	9	9	9 (100%)	0 (0%)	0 (0%)
	I	0	0	0	0	0	0			
RCC	S	22	22	22	22	22	22	22 (100%)	0 (0%)	0 (0%)
	I	0	0	0	0	0	0			

**Table 3 genes-12-00197-t003:** STR and MSI status of all investigated MSI positive samples (*n* = 19).

Sample ID	Tumor Type	Alteration (%)		Status
		STR	MSI	
138	CRC	84.21%	100.00%	MSI-H
166	CRC	55.56%	83.33%	MSI-H
180	CRC	68.42%	100.00%	MSI-H
200	CRC	31.58%	16.67%	MSI-L
214	CRC	52.63%	83.33%	MSI-H
220	CRC	78.95%	100.00%	MSI-H
230	CRC	42.11%	66.67%	MSI-H
237	GC	78.95%	100.00%	MSI-H
244	GC	26.32%	33.33%	MSI-H
245	GC	47.37%	16.67%	MSI-L
247	GC	36.84%	100.00%	MSI-H
272	GC	73.68%	100.00%	MSI-H
286	GC	52.63%	100.00%	MSI-H
300	GC	68.42%	100.00%	MSI-H
302	GC	52.63%	16.67%	MSI-L
307	GC	36.84%	100.00%	MSI-H
312	GC	73.68%	16.67%	MSI-L
328	GC	31.58%	83.33%	MSI-H
337	GC	68.42%	100.00%	MSI-H

**Table 4 genes-12-00197-t004:** The percentage of STR positive samples in different cutoff values. The bold fonts refer to the most suitable cutoff value for the STR alterations; Perc. refers to the STR alteration frequency.

Tumor Type	No. of Total Samples	Cutoff of STR Alterations												ORR with Anti-PD1 or Anti-PD-L1 [7]
		≥5.26% (1 Out of 19)		≥10.53% (2 Out of 19)		≥15.79% (3 Out of 19)		≥21.05% (4 Out of 19)		≥26.32% (5 Out of 19)		≥31.58% (6 Out of 19)		
		Sample No.	Perc.	Sample No.	Perc.	Sample No.	Perc.	Sample No.	Perc.	Sample No.	Perc.	Sample No.	Perc.	
BC	36	13	36.11%	5	13.89%	4	11.11%	4	11.11%	4	11.11%	1	2.78%	~5%
HCC	30	19	63.33%	15	50.00%	11	36.67%	8	26.67%	6	**20.00%**	3	10.00%	~17%
PC	35	9	25.71%	5	14.29%	3	8.57%	3	8.57%	3	8.57%	2	5.71%	~0%
CRC	129	90	69.77%	80	62.02%	67	51.94%	51	39.53%	44	**34.11%**	24	18.60%	~30%
GC	121	66	54.55%	48	39.67%	42	34.71%	37	30.58%	36	**29.75%**	30	24.79%	~30%
LC	25	12	48.00%	12	48.00%	10	40.00%	8	32.00%	6	**24.00%**	6	24.00%	~21%
EC	9	7	77.78%	7	77.78%	5	55.56%	5	55.56%	4	44.44%	4	44.44%	22–23.5%
RCC	22	9	40.91%	4	**18.18%**	3	13.64%	2	9.09%	2	9.09%	1	4.55%	~25%
**Tumor Type**	**No. of Total Samples**	**Cutoff of STR Alterations**												**ORR with Anti-PD1 or Anti-PD-L1** [7]
		**≥36.84% (7 Out of 19)**		**≥42.11% (8 Out of 19)**		**≥47.37% (9 Out of 19)**		**≥52.63% (10 Out of 19)**		**≥57.89% (11 Out of 19)**		**☐**		
		**Sample No.**	**Perc.**	**Sample No.**	**Perc.**	**Sample No.**	**Perc.**	**Sample No.**	**Perc.**	**Sample No.**	**Perc.**	**☐**	**☐**	
BC	36	1	2.78%	1	2.78%	1	2.78%	1	2.78%	0	0.00%	☐	☐	~5%
HCC	30	0	0.00%	0	0.00%	0	0.00%	0	0.00%	0	0.00%	☐	☐	~17%
PC	35	2	5.71%	2	5.71%	2	5.71%	2	5.71%	1	2.86%	☐	☐	~0%
CRC	129	17	13.18%	11	8.53%	8	6.20%	5	3.88%	3	2.33%	☐	☐	~30%
GC	121	21	17.36%	17	14.05%	13	10.74%	9	7.44%	7	5.79%	☐	☐	~30%
LC	25	4	16.00%	2	8.00%	2	8.00%	1	4.00%	1	4.00%	☐	☐	~21%
EC	9	4	44.44%	3	33.33%	2	**22.22%**	2	22.22%	1	11.11%	☐	☐	22–23.5%
RCC	22	1	4.55%	1	4.55%	1	4.55%	1	4.55%	0	0.00%	☐	☐	~25%

**☐,** It was the empty colomns.

## Data Availability

None.

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
