# Peer review of "Investigation of an Alternative Marker for Hypermutability Evaluation in Different Tumors"

_genes, 2021, doi:10.3390/genes12020197_

Round 1

Reviewer 1 Report

Chen et al evaluated the ability of a previously developed method (doi: 10.1007/s11684-019- 298 0698-4) for detecting MSI, on tumor types not typically associated with microsatellite instability. While it is important to detect MSI on many other tumor types due to their response to immune checkpoint inhibitors, Chen et al failed to cite other attempts along these lines in spite of the fact that many attempts were published. 

While this is a technical issue that can be easily improved, the main problem that I found regarding their article is that they tested many other tumor types, each are known to have around 0.5%-2% MSI cases. Their sample size from each tumor type is too small in order to enable detection in such low levels thus the fact that they haven't found MSI in tumor types like Lung and more does not indicate that they are not there. It just says that this is below their detection threshold. Moreover, they checked specific loci, and these tumor types may have mutations in other loci. 
As a conclusion the attempt to validate their previously described method on other tumor types is important, but their analysis suffers from sample size issues that should de addressed. Just to clarify this is a major issue, as the whole goal was to analyze MSI in other tumor types besides colon, stomach, and endometrial, but it looks as if they were not able to do it. 

Other comments:

  1. They should describe their method from doi: 10.1007/s11684-019- 298 0698-4 in this current manuscript, at least breifly. 
  2. The writing style should be improved, e.g. tenses should be consistant. 
  3. Sometimes it is not clear if they are trying yo measure miscrostallite instability or TMB in general. 

Author Response

Dear Reviewer,

We appreciate the time and effort the you have made in reviewing our manuscript entitled “Investigation of an alternative marker for hypermutability evaluation in different tumors” (Manuscript ID: genes-1055149).

We are very glad to receive so many thoughtful comments, which are very helpful for us improving the manuscript. We have studied the comments very carefully, and all the corrections were marked in red in the revision. The responses to the reviewers’ comments are as following:

Reviewer 1#:

Chen et al evaluated the ability of a previously developed method (doi: 10.1007/s11684-019- 298 0698-4) for detecting MSI, on tumor types not typically associated with microsatellite instability. While it is important to detect MSI on many other tumor types due to their response to immune checkpoint inhibitors, Chen et al failed to cite other attempts along these lines in spite of the fact that many attempts were published.

While this is a technical issue that can be easily improved, the main problem that I found regarding their article is that they tested many other tumor types, each are known to have around 0.5%-2% MSI cases. Their sample size from each tumor type is too small in order to enable detection in such low levels thus the fact that they haven't found MSI in tumor types like Lung and more does not indicate that they are not there. It just says that this is below their detection threshold. Moreover, they checked specific loci, and these tumor types may have mutations in other loci.

As a conclusion the attempt to validate their previously described method on other tumor types is important, but their analysis suffers from sample size issues that should de addressed. Just to clarify this is a major issue, as the whole goal was to analyze MSI in other tumor types besides colon, stomach, and endometrial, but it looks as if they were not able to do it.

We would like to thank the reviewer’s careful reading and valuable comments. According to the reviewer’s suggestion, we cited the methods of MSI-H/MMR-D screening in both colorectal and non-colorectal tumors (line 54-58).

The patients suffering from gastrointestinal cancers have more surgeon opportunities in clinic. In addition, the gastrointestinal cancers held well-defined clinical implications in MSI testing, and it was sample types we focused. That was the reason why the sample types like esophageal cancer was comparative small. It was reported that the dMMR/MSI-H in esophageal cancer was 0–3.3% (Therap Adv Gastroenterol. 2020; 13: 1756284820948773.), and we also did not observe the MSI positive samples in esophageal cancer. The results we released was basically consistent with the facts. Of course, we cannot deny the low sample size of esophageal cancer and lung cancer, and larger sample size would compromise stronger evidences. We quite agreed with the reviewer, that the limitation on sample size should be addressed. We would like to depict the issue of sample size in the revision, and the further studies would follow up (line 287-290). We kept contact with the hospitals, and was trying to expand the sample types. Our goals for the future is to enlarge the sample size and tumor types.

Nearly all panels would be designed for specific loci regardless of WGS. The MSI panel was a commercial kit designed for MSI detection, and the markers have been widely used and validated worldwide. In the present study, the MSI detection kit was regarded as benchmarking or golden standard, and the detection effectiveness of STR was what we would like to discuss. Not all tumor groups harbored MSI positive samples, however, all of them harbored varied degree of STR alternations. It demonstrated that the STR markers might be an alternative marker for hypermutability screening better than microsatellite markers.

Other comments:

  1. They should describe their method from doi: 10.1007/s11684-019- 298 0698-4 in this current manuscript, at least breifly.

We would like to thank the reviewer for the valuable suggestion. The brief method description had been introduced in the revision (line 107-111).

  1. The writing style should be improved, e.g. tenses should be consistant.

We would like to thank the reviewer for the careful reading, and felt sorry for our carelessness. In the revision, we proofread the manuscript carefully and improve the tenses with the help of English Language Editing Service.

  1. Sometimes it is not clear if they are trying to measure miscrostallite instability or TMB in general.

We would like thank the reviewer for the comments. The MSI status must be measured because it was the current standard. Due to the narrow range and sensitivity provided by MSI panels, more sensitive markers should be discovered. It was exactly as the reviewer mentioned, that many other attempts were tried to increase the sensitivity. And TMB was one of approach which had been widely used along with the popularization of NGS. Therefore, a comparison between TMB and STR had been made using few lines of description (line 192-196). A similar trend of mutability was observed across the tumor types, and higher value was detected using STR. This result demonstrated that STR might be a more sensitive predictor of hypermutability.

In general, we would like to introduce STR, as an alternative biomarker for hypermutability evaluation. Both the MSI and TMB were the competitive benchmarking of STR.

We appreciate for your warm work, and hope you will be satisfied with the corrections. Thank you again for your good comments.

Sincerely yours,

Anqi Chen

Reviewer 2 Report

Chen et al., report a comparative assessment of forensic short tandem repeats (STR) markers vs microsatellite instability (MSI) assay panels for better evaluation of hypermutability and thus tumor mutation burden (TMB). Briefly, they reason that the use of limited number of MSI markers for measuring TMB undercounts the patient population who could therapeutically benefit from immunotherapy application. To this end, they assess alteration frequency of a panel of STR markers and a panel of commonly used MSI markers across eight different cancer types using standardized assays. STR markers are rarely used outside of forensic applications. The authors report of their usability as a better predictor for success in immunotherapy, citing greater range and sensitivity across multiple cancer types compared to MSI panels.

Accurate assessment of TMB is of significant interest to the field of immunotherapy and thus underscores the relevance of the manuscript to a broader audience. Overall, the manuscript is reasonably well written but could be re-worked to improve clarity (both grammatical and communication) at many instances. My concerns are described below:

  1. The explanation for acronyms used in multiple places should be consistent throughout the manuscript. For e.g. colon cancer is CRC (line 21) but mentioned to be colorectal cancer on line 55. Between the abstract and the end of introduction they have been repeated three times. This could be easily avoided to improve readability.
  1. On line 40, elevated microsatellite alterations are just not observed at EMAST (as shown by the authors themselves) but also at more commonly used mono and di-nucleotide repeats. This could be added.
  1. On line 44, there is a typo in the word “g,,,,,,l”
  1. Line 86, MSS refers…….stable microsatellites, not MSI
  1. The description for “S” in figure 2B is missing.
  1. Define what is considered MSI-high and MSI-low in your analysis.
  1. Could the inclusion of a greater number of STR markers (19) for analysis compared to just 8 for MSI explain the higher alteration frequency being detected in STRs? The authors have repeatedly highlighted that low number of MSI markers in regular MSI analysis panel could explain a lack of sensitivity. A discussion on this issue will provide clarity.
  1. Although for the most part, the observations concerning MSI alteration % reported in the manuscript agree with the published literature (PMID: 27694933), the total absence of MSI events in endometrial cancer (EC) group is a quite surprising. This could presumably be explained by acknowledging a very low sample size compared to gastric cancer (GC) or colorectal cancer (CRC). However, the STR alteration frequencies for EC are the highest in the group. This highlights the need for an assessment of variation in sample size affecting/limiting meaningful comparison. This should be addressed or acknowledged. It would also be helpful to have a depiction of how close (or not) the data match between MSI alteration % and STR alteration % across the samples.
  1. The legend for table 2 lacks an explanation for sample numbers shown for each group.
  1. Line 154, “STR contributed…………..the MSI”. The word “contribution” needs re-assessment as it was just detected to harbor/reveal more alterations compared to MSI.
  1. Line 161 and 162: “TMB…………………..expression”. This is misquoted. PD-L1 expression and TMB are independent predictors of response to immune checkpoint therapy. This should be corrected.
  1. Line 233 has a typo in “expanse”.
  1. Line 244 needs a reference.
  1. Table 4 and its explanation in the text needs to be clarified to a great extent to be decipherable. It is not clear to me how the most suitable cutoff values (shown in bold) were determined. I found this to be the most difficult part of the paper to read and comprehend.

Author Response

Dear reviewer,

We appreciate the time and effort the you have made in reviewing our manuscript entitled “Investigation of an alternative marker for hypermutability evaluation in different tumors” (Manuscript ID: genes-1055149).

We are very glad to receive so many thoughtful comments, which are very helpful for us improving the manuscript. We have studied the comments very carefully, and all the corrections were marked in red in the revision. The responses to the reviewers’ comments are as following:

Reviewer 2#

Chen et al., report a comparative assessment of forensic short tandem repeats (STR) markers vs microsatellite instability (MSI) assay panels for better evaluation of hypermutability and thus tumor mutation burden (TMB). Briefly, they reason that the use of limited number of MSI markers for measuring TMB undercounts the patient population who could therapeutically benefit from immunotherapy application. To this end, they assess alteration frequency of a panel of STR markers and a panel of commonly used MSI markers across eight different cancer types using standardized assays. STR markers are rarely used outside of forensic applications. The authors report of their usability as a better predictor for success in immunotherapy, citing greater range and sensitivity across multiple cancer types compared to MSI panels.

Accurate assessment of TMB is of significant interest to the field of immunotherapy and thus underscores the relevance of the manuscript to a broader audience. Overall, the manuscript is reasonably well written but could be re-worked to improve clarity (both grammatical and communication) at many instances. My concerns are described below:

1. The explanation for acronyms used in multiple places should be consistent throughout the manuscript. For e.g. colon cancer is CRC (line 21) but mentioned to be colorectal cancer on line 55. Between the abstract and the end of introduction they have been repeated three times. This could be easily avoided to improve readability.

We felt sorry for our carelessness, and would like to thank the reviewer for the careful reading. The mistake of inconsistent statements had been corrected in the revision (line 22). At the same time, the repeated statements were also corrected (line 64).

We really appreciate the reviewer’s nice suggestion. To improve the readability, we proofread the manuscript carefully and grammatically improved with the help of English Language Editing Service.

2. On line 40, elevated microsatellite alterations are just not observed at EMAST (as shown by the authors themselves) but also at more commonly used mono and di-nucleotide repeats. This could be added.

We appreciated the reviewer’s rigorous attitude, and it was our fault ignoring the outcome produced by mono and di-nucleotide repeats. The statement of “the commonly used mono and di-nucleotide repeats [3]” was added according to the reviewer’s suggestion (line 41-42).

3. On line 44, there is a typo in the word “g,,,,,,l”

We were sorry for our carelessness, and would like thank the reviewer for the careful reading. The typo of “geastroinestinal” was corrected to “gastrointestinal” in the revision (line 47).

4.Line 86, MSS refers…….stable microsatellites, not MSI

Thanks to the reviewer’s comments, the wrong statement of MSS was corrected in the revision. The statement of “MSI” was corrected as “stable microsatellites” (line 96).

5.The description for “S” in figure 2B is missing.

According to the reviewer’s nice suggestion, and the description for “S” in figure 2B was added in the revision (line 133).

6. Define what is considered MSI-high and MSI-low in your analysis.

The MSI status was defined as the number of instable microsatellite markers. The MSI-high was defined once more than two instable microsatellites observed, and the MSI-low was defined only one instable microsatellites detected. The detailed evaluation of MSI-high and MSI-low had been described in the section of materials and methods (line 94-96).

7. Could the inclusion of a greater number of STR markers (19) for analysis compared to just 8 for MSI explain the higher alteration frequency being detected in STRs? The authors have repeatedly highlighted that low number of MSI markers in regular MSI analysis panel could explain a lack of sensitivity. A discussion on this issue will provide clarity.

We would like to thank the reviewer for the comments. We thought more markers might produce higher chance detecting the mutations. Based on the 6-MSI system, the patient might be harbor the mutation of 16.67% (1/6), 33.33% (2/6), 50% (3/6) or above. Since MSI-H and MSI-L was defined as the patient who harbored one and more than one alternation respectively. Therefore, the interpretation for the patients harboring 16.67%~33.33% alternation was ambiguous. The discussion on the disadvantage of low number of MSI markers had been included in the revision (line 250-254).

8. Although for the most part, the observations concerning MSI alteration % reported in the manuscript agree with the published literature, the total absence of MSI events in endometrial cancer (EC) group is a quite surprising. This could presumably be explained by acknowledging a very low sample size compared to gastric cancer (GC) or colorectal cancer (CRC). However, the STR alteration frequencies for EC are the highest in the group. This highlights the need for an assessment of variation in sample size affecting/limiting meaningful comparison. This should be addressed or acknowledged. It would also be helpful to have a depiction of how close (or not) the data match between MSI alteration % and STR alteration % across the samples.

The patients suffering from gastrointestinal cancers have more surgeon opportunities in clinic. In addition, the gastrointestinal cancers held well-defined clinical implications in MSI testing, and it was sample types we focus most. That was the reason why the sample types like EC was comparative small. It was reported that the dMMR/MSI-H in EC was 0–3.3% (Therap Adv Gastroenterol. 2020; 13: 1756284820948773.), and we observed the high STR alternation rate and absence of MSI positive samples in EC. The result of EC might be of the real situation, at the same time, it might be a biased result attribute to the comparative low sample size. The results we released was basically consistent with the facts. Of course, we cannot deny the low sample size of esophageal cancer, and larger sample size would compromise stronger evidences. According to the reviewer’s nice suggestion, we would like to depict the issue of sample size in the revision, and the further studies would follow up (line 287-290). We kept contact with the hospitals, and was trying to expand the sample types. Our goals for the future is to enlarge the sample size and tumor types.

The STR alteration rates of all MSI positive samples were shown in Table 3, and we found that all MSI-H samples possessed varied degrees of STR mutations. However, the value of MSI and STR was not strictly matched due to the different detection range between the two panels.

9. The legend for table 2 lacks an explanation for sample numbers shown for each group.

According to the reviewer’s nice request, we would like to explain the sample number for each group in the legend for table 2 (Table 2, line 164-165).

10. Line 154, “STR contributed…………..the MSI”. The word “contribution” needs re-assessment as it was just detected to harbor/reveal more alterations compared to MSI.

We appreciated the reviewer’s suggestion, and felt sorry for the improper statement. The statement of “reveal” was much more precise than “contribute”. Our mistake was corrected in the revision, and the statement was corrected as “reveal” in line 117 and 173.

11. Line 161 and 162: “TMB…………………..expression”. This is misquoted. PD-L1 expression and TMB are independent predictors of response to immune checkpoint therapy. This should be corrected.

The improper statement was modified thanks to the reviewer’s careful reading. The statement of “TMB……….expression” was corrected as “TMB has been widely recognized as the biomarker for response to immunotherapy” (line 182).

12. Line 233 has a typo in “expanse”.

We felt sorry for the typo, and would like thank the reviewer for the comments. The mistake in line 259 had been corrected in the revision.

13. Line 244 needs a reference.

We would like to thank the reviewer for the nice suggestion, and the reference had been provided in the revision (line 271).

14. Table 4 and its explanation in the text needs to be clarified to a great extent to be decipherable. It is not clear to me how the most suitable cutoff values (shown in bold) were determined. I found this to be the most difficult part of the paper to read and comprehend.

Although the STR testing revealed its potential ability in predicting the response in to immunotherapy. However, we still did not know to what extent (or cut-off of STR alternation rate) could we defined a person as STR-instablility. Table 4 would like try to mimic the potential threshold for predicting STR-instability based on the corresponding ORR with anti-PD1 or anti-PD-L1. There were 19 STR loci included in the panel, which should provide the STR alternation rate from 5,26% (1/19) ~ 100% (19/19). To take the CRC for instance, the ORR was 30%. Therefore, there should be 39 (30% * 129 =38.7) patients should response to the immunotherapy. Next, we list the number of the patients in different cutoffs (harboring 5.26% STR alternation to 57.89% STR alternation). We found that there were 44 patients in the cutoff the 26.32%, which was the closest to its theoretical value (n=39). Thereby, the cutoff of 26.32% was settled for CRC. In addition, all the MSI-H samples harbored at least 26.32% STR alternation. Therefore, we predicted the cutoff for CRC should be 26.32%.

In general, the Table 4 was the predicted cutoff of STR-instability based on the published ORR, and the real response rate should be validated with the clinical trials.

We appreciate for your warm work, and hope you will be satisfied with the corrections. Thank you again for your good comments.

Sincerely yours,

Anqi Chen
